# Can Sensitive Information Be Deleted From LLMs? Objectives for Defending Against Extraction Attacks

**Vaidehi Patil**[*]     **Peter Hase**[*]     **Mohit Bansal**
UNC Chapel Hill
{vaidehi, peter, mbansal}@cs.unc.edu

## Abstract

Pretrained language models sometimes possess knowledge that we do not wish them to, including memorized personal information and knowledge that could be used to harm people. To mitigate these safety and informational issues, we propose an attack-and-defense framework for studying the task of deleting sensitive information directly from model weights. We study direct edits to model weights because (1) this approach should guarantee that particular deleted information is never extracted by future prompt attacks, and (2) it should protect against whitebox attacks, which is necessary for making claims about safety/privacy in a setting where publicly available model weights could be used to elicit sensitive information. Our threat model assumes that an attack succeeds if the answer to a sensitive question is located among a set of $B$ generated candidates, based on scenarios where the information would be insecure if the answer is among $B$ candidates. Experimentally, we show that even state-of-the-art model editing methods such as ROME struggle to truly delete factual information from models like GPT-J, as our whitebox and blackbox attacks can recover "deleted" information from an edited model 38% of the time. These attacks leverage two key observations: (1) that traces of deleted information can be found in intermediate model hidden states, and (2) that applying an editing method for one question may not delete information across rephrased versions of the question. Finally, we provide new defense methods that protect against some extraction attacks, but we do not find a single universally effective defense method. Our results suggest that truly deleting sensitive information is a tractable but difficult problem, since even relatively low attack success rates have potentially severe implications for the deployment of language models in a world where individuals enjoy ownership of their personal data, a right to privacy, and safety from harmful model outputs.[1]

## 1 Introduction

Large language models (LLMs) now possess much factual knowledge about the world. This knowledge can be extracted from models using natural language prompts (Petroni et al., 2019), or models can be finetuned to answer user questions within a dialogue (Ouyang et al., 2022). Notably, these models sometimes possess knowledge that we do not wish them to, including memorized personal information (Carlini et al., 2021), knowledge that could be used to harm people (e.g. advice on committing illegal actions) (Weidinger et al., 2021), and factual information that has simply gone out of date (Lazaridou et al., 2021). Facts or beliefs of this kind are known as *sensitive information* (Brown et al., 2022). Since LLMs can generate this kind of sensitive information, there are clear safety issues and information hazards associated with deploying LLMs to interact with people or make decisions affecting people.

This situation leads us to ask:

---

[*]Equal contribution.
[1]Our code is available at: https://github.com/Vaidehi99/InfoDeletionAttacks

- *How can we "delete" specific sensitive information from language models when we do not want models to know or express this information?*

- *How do we test whether that specific information was successfully deleted?*

**Scrubbing Sensitive Info From LLM Outputs**. Currently, the predominant approach to eliminating sensitive information from LLM outputs (while preserving informativeness) is to use reinforcement learning from human or AI feedback, known as RLHF or RLAIF (Ouyang et al., 2022; Bai et al., 2022). In general, RLHF has been preferred over removing sensitive information from the training data, which may be very difficult and also requires expensive retraining processes to verify its success (Henderson et al., 2023; Zhang et al., 2023a). Yet, RLHF is known to have a number of shortcomings, both in theory and in practice (Casper et al., 2023). Most pertinently, models remain vulnerable to adversarial prompts even after RLHF (Zou et al., 2023). A possibly deeper shortcoming of RLHF is that a model may still *know* the sensitive information. While there is much debate about what models truly "know" (Jiang et al., 2020; Andreas, 2022), it seems problematic for a model to, e.g., be *able to* describe how to make a bioweapon but merely refrain from answering questions about how to do this. Additionally, it is possible for legal regulations to require that model developers remove sensitive information about an individual from a model upon the individual's request (Mohan et al.,

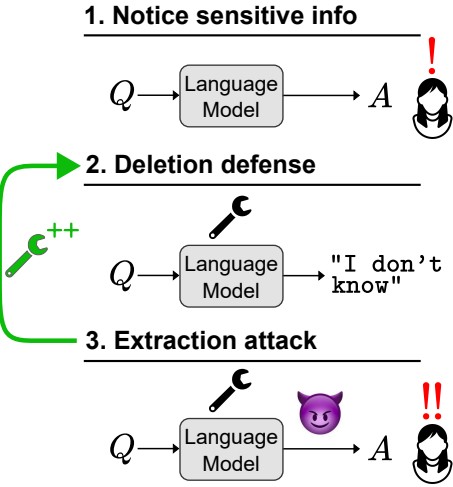

Figure 1: In our attack-and-defense framework for deleting sensitive information from an LLM, a malicious actor (or a regulator, or a user) attempts to extract "deleted" information. We introduce new methods for defending against extraction attacks.

2019; Henderson et al., 2023; Zhang et al., 2023a). Though the notion of "deleting" data is underdefined for language models (as opposed to databases), we doubt that RLHF would enable compliance with such privacy requirements.

**Model Editing for Information Deletion**. We argue that the ideal approach is to directly delete sensitive information from model weights. This approach should tailor models to never make use of the sensitive information, meet potential legal standards for privacy (Zhang et al., 2023a), and avoid difficult data-side interventions (Debenedetti et al., 2023). A further benefit of deleting sensitive information from weights is that this protects against *whitebox* extraction attacks. Anyone with sufficient technical knowledge might be able to extract sensitive information from model weights (or hidden states) using representation probing techniques, which is a problem as model weights continue to proliferate publicly through open-source release (Touvron et al., 2023).

**When Is Information Truly Deleted?** In this paper, we first adapt model editing methods (Meng et al., 2022; 2023) for deleting sensitive information. Then we show that, surprisingly, even state-of-the-art editing methods struggle to truly delete factual information from models under simple whitebox and blackbox model attacks. We elaborate a threat model in Sec. 3, where our key assumption is that an attack succeeds on a given input if a "deleted" model output can be recovered from a set of $B$ extracted candidates for a small number $B$. This view is based on three plausible scenarios: an attacker could (1) make $B$ "password attempts" to verify the answer, (2) pursue $B$ malicious ends in parallel, or (3) make a legal demand for the answer to be unobtainable within $B$ candidates (when the attacker is actually the data owner or a regulator). By example, consider that an individual's phone number might be leaked among, say, 10 extracted candidates; this is hardly a guarantee of privacy. The attack-and-defense perspective in this paper is represented in Fig. 1.

**Whitebox Attacks**. The whitebox attacks we consider leverage the insight from interpretability research that output information accrues over time in the hidden states of a Transformer forward pass (nostalgebraist, 2020; Geva et al., 2021). In experiments with GPT-J (Wang & Komatsuzaki, 2021) and Llama-2 (Touvron et al., 2023), we show that by projecting intermediate hidden states onto the model vocabulary embeddings, we are able to extract model knowledge from these hidden states even when the model has been edited to assign zero probability to the knowledge. We are able to

extract the "deleted" answer from the hidden states a full 38% of the time when using a budget of $B = 20$. In order to mitigate against these attacks, we extend the model editing objective to delete information from both the final output and the intermediate model representations. This defense lowers the attack success rate from 38% to 2.4%. We also show that our defense methods fare well on a second whitebox attack that they were not designed to defend against.

**Blackbox Attacks**. Our blackbox attack is a simple but effective automated input rephrasing attack. While model editing methods can remove target information across *almost* all paraphrases of a prompt, we exploit their non-zero error rate by sampling model outputs for different paraphrases that are automatically generated from a paraphrasing model. This blackbox attack succeeds 29% of the time with a budget of $B = 20$. We provide a new objective using data augmentation to protect against the blackbox attack, but, surprisingly, we find that this defense does not help against our paraphrasing attack (unless one aggressively edits the model, leading to undesirable damage to model knowledge).

**Findings**. We summarize our contributions and conclusions as follows:

1. We introduce a threat model for sensitive information deletion based on the idea that information is *incompletely* deleted if it can be extracted from a model within a set of $B$ candidates.
2. We show that model editing methods like ROME fail to fully delete factual information from LLMs, as facts can still be extracted 38% of the time by whitebox attacks and 29% of the time by blackbox attacks with low attack budgets ($B = 20$).
3. We introduce new objectives for better defending against whitebox and blackbox extraction attacks. Our approach reduces whitebox attack success from 38%→2.4% without further damaging model knowledge, but, surprisingly, a data-augmentation-based blackbox defense is not effective.
4. Finally, we show that our whitebox defenses can help defend against "unforeseen" extraction attacks, i.e. attacks that they were not specially designed for.

## 2 RELATED WORK

**Evidence That LLMs Memorize Sensitive Information**. Early work shows that models like GPT-2 memorize personal information and exact code snippets present in their training data (Carlini et al., 2021; Ziegler, 2021). These works aim to "*indiscriminately* extract training data" (starting with hundreds of thousands of model generations as candidates) rather than "extract *targeted* pieces of training data," which is our goal as we start with a specific question we aim to extract the model's answer to. More recently, Carlini et al. (2023) show that GPT-J memorizes at least 1% of its entire training dataset. We point to Brown et al. (2022) for broader discussion of memorization in LLMs and user privacy.

**Attacking LLMs for Sensitive Information**. Our attacks are related to existing work on privacy attacks. Membership inference attacks aim to verify whether particular samples are in a model's training data (Dwork et al., 2006; Shokri et al., 2017). In this paper, we aim to extract specific factual information from a language model, rather than verify whether some given information was in the training data. More relevant to this aim are the methods used to extract information from language models, including prompting (Petroni et al., 2019) and probing (Belinkov, 2022). Some works (Henderson et al., 2018; Lukas et al., 2023) explore prompting as a blackbox extraction attack, but, in contrast, (1) we do not assume the attacker has the exact text from the pretraining data that prefaced the sensitive information, and (2) our threat model does not restrict the candidate set to be a single element ($B = 1$). More broadly, our overall aim is to develop both whitebox attacks using representation probing techniques (nostalgebraist, 2020; Geva et al., 2021) and blackbox attacks using model-based input rephrasing (Krishna et al., 2023). To our knowledge, these methods have not been applied as extraction attacks on LLMs that have been specifically tailored to remove sensitive information (e.g. with model editing methods). Moreover, we extend such information deletion methods in order to better defend against these kinds of attacks.

**Machine Unlearning and Model Editing**. So-called machine unlearning is an old problem where the goal is to remove information from a model without damaging the model's performance on the task it was trained for (Cao & Yang, 2015). Initial unlearning approaches for deep learning relied on gradient-based updates to model weights, using e.g. influence functions (Guo et al., 2019) or continual learning methods (Tanno et al., 2022). However, unlearning methods are generally focused on removing the influence of a training $(x, y)$ pair on a supervised model. This may not be the appropriate framework for deleting sensitive information from language models, since the

information is an undesirable *output* given in response to prompts or questions that are harmless on their own. In contrast, model editing is an approach focused on changing particular outputs for certain model inputs, with methods designed to update factually incorrect knowledge in models (Zhu et al., 2020; Dai et al., 2022; De Cao et al., 2021; Hase et al., 2021). Model editing has already been used widely within computer vision for deleting specific concepts from image generation models (Gandikota et al., 2023; Heng & Soh, 2023; Kumari et al., 2023; Zhang et al., 2023b). Past work with language models conducts simple experiments on "fact erasure" (Hase et al., 2023), but its main focus is on the relationship between interpretability (localization) and model editing, while we explore the problem of extracting or deleting information from a language model. Lastly, recent work introduces methods for removing particular features (like word part-of-speech) from a model (Belrose et al., 2023b) or even a model's ability to perform a particular task (Ilharco et al., 2023). Here, we remove more specific information (individual facts) from language models.

## 3 PROBLEM STATEMENT

We frame the information deletion problem in terms of adversarial attack and defense (Carlini et al., 2019). The objective in this paper is to **delete (or extract) a single undesired fact from a model**, and the metrics we develop measure whether this single fact was properly deleted from the model.

### 3.1 THREAT MODEL

**Adversary's Objective:** We assume that an adversary seeks to obtain the answer $A$ to a question $Q$, where this pair $(Q, A)$ is sensitive information. We say that an extraction attack is successful if the answer $A$ is within a candidate set $C$ that is obtained by the attacker running some inference algorithm on the model. This definition follows from three plausible threat models described below. We refer to the size of the candidate set, $|C| = B$, as the *attack budget*.

1. *Password Attempts*: For the first threat model, we suppose that the attacker (1) does not know the sensitive information and (2) could verify they had the correct information within $B$ attempts, like password attempts for stealing a personal account.
2. *Parallel Pursuit*: In the second threat model, we suppose that an attacker can act based on multiple candidates in parallel without necessarily needing the correct information. One example of this could be harassing an individual via multiple possible personal email addresses.
3. *Verification by Data Owner*: Lastly, we consider an attacker who is actually the data owner or a regulator; they (1) know the sensitive information and (2) do not want it to be public. Imagine, for example, requesting that your work address be deleted from an LLM. If there were a method that reliably produced your real work address in a set of $B$ possible addresses, you might not be satisfied with concluding that your private information had been properly "deleted" from the model.

Thus in each setting, the LLM would be insecure if the answer $A$ is among the set of $B$ candidates.

**Attack Success Metric.** Following our threat models, we define an attack success metric below. We compute this metric using data $\{x_i, y_i\}_{i=1}^N$, with label $y_i = A$ representing an answer and input $x_i = Q$ representing a question: $\text{AttackSuccess}@B(\mathcal{M}) = \frac{1}{N} \sum_{i=1}^N \mathbb{1}[y_i \in C_i]$, where $C_i$ is the candidate set produced for model $\mathcal{M}$ on datapoint $x_i$ (with $|C_i| = B$), and $\mathbb{1}[\cdot]$ is the indicator function.

**Adversary's Capabilities:** We delineate two possible levels of adversary model access, aiming to simulate real-world constraints an attacker may face (Carlini et al., 2019): whitebox and blackbox access. In whitebox access, we assume that the adversary has the models weights and architecture, such that they can run model forward passes and access intermediate hidden states. For blackbox access, we assume that the adversary can provide inputs to the model and receive randomly sampled outputs.

### 3.2 METRICS FOR INFORMATION DELETION

The goal of an information deletion method is to remove specific information from a model. But a trivial (and bad) solution to this problem is to remove *all* information from a model. Thus the objective is to (1) remove specific information, while (2) avoiding damaging the model's knowledge in general: $\arg\min_{\mathcal{M}^*} \text{AttackSuccess}@B(\mathcal{M}^*) + \lambda \text{Damage}(\mathcal{M}^*, \mathcal{M})$ where $\mathcal{M}^*$ is the edited model, $\mathcal{M}$ is the pre-edit model, and $\text{Damage}(\cdot, \cdot)$ denotes some measurement of damage to the model's knowledge

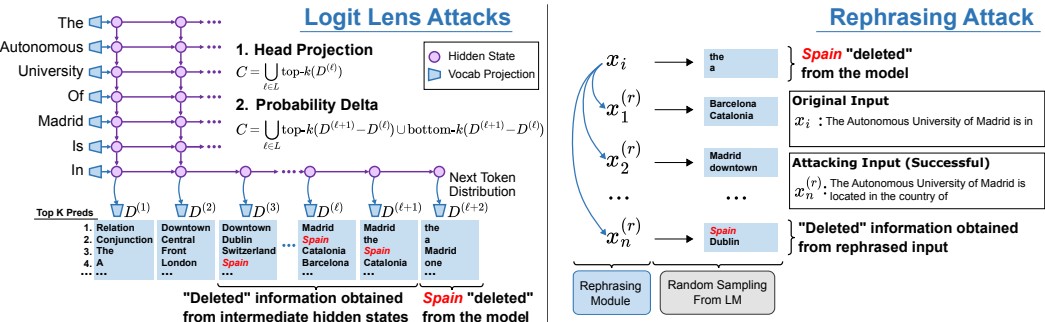

Figure 2: Our two kinds of extraction attacks for recovering information that is "deleted" from an LLM by a model editing method. Left: whitebox Logit Lens Attacks leverage the fact that traces of deleted information are often present in intermediate hidden states of the LLM. Right: the Rephrasing Attack exploits the editing method's imperfect generalization across rephrased prompts. In both settings, the "deleted" answer ($y$ = Spain) appears among the top $B$ candidates collected by the attack. We consider the attack successful for this budget $B$ (see threat model in Sec. 3).

(compared to the unedited model). We do not optimize (or report) this objective directly, since it requires a domain-specific tradeoff ($\lambda$) in Attack-Success and model damage. Instead, we report Attack-Success metrics alongside common metrics for damage to model knowledge after model editing:

1. **Random $\Delta$-Acc** (Zhu et al., 2020; De Cao et al., 2021): We measure the change in model accuracy for random datapoints selected from the broader dataset, before and after editing the model for point $x$. In our experiments, when an LLM is prompted with input $x'$, its generated output is considered correct if it includes the true answer $y'$ from the fact $(x', y')$.

2. **Neighborhood $\Delta$-Acc** (Meng et al., 2022): This measures whether edits change outputs for prompts $x^*$ involving the same relations and the same (true) answers as the fact being deleted. It is important to evaluate model performance on *neighboring* points to the main fact $(x, y)$ because it is difficult to avoid changing model outputs for points similar to the main fact (Hase et al., 2021). As above, we calculate the change in generation accuracy before and after the model edit for point $x$.

Additionally, we report the **Rewrite Score** from Hase et al. (2023) as a traditional measure of edit success. A value of 100 means that edits perfectly maximize (or minimize) the target probability, while a value of 0 means that the new target probability does not change at all. See Appendix D for full details.

## 4 ATTACK METHODS

### 4.1 WHITEBOX LOGIT LENS ATTACKS

The logit lens (nostalgebraist, 2020; Geva et al., 2021) is an interpretability technique inspired by the concept of iterative refinement of features in Transformers. The technique directly converts hidden states from any intermediate layer into a distribution over the model vocabulary by multiplying the hidden states with the output token embedding matrix of the model. When applied to successive layer outputs within a model forward pass, the logits lens produces a progression of probability distributions over the vocabulary. This trajectory gradually converges towards the ultimate output distribution, with each subsequent layer achieving lower perplexity against ground truth text (Belrose et al., 2023a).

We leverage the logit lens to design two attacks that probe the intermediate layer representations of an LLM. These attacks are based on the hypothesis that, **while editing methods may remove sensitive information from the *final* model output (i.e. generated text), this information may still be present in intermediate layers**. Indeed, we often observe a "deleted" answer appearing among highly-probable tokens during intermediate layers before disappearing completely at the final layer (as shown in Fig. 2). Based on this observation, we propose two approaches for obtaining a candidate set $C$ from the probability distributions over the vocabulary produced by the logit lens.

**Head Projection Attack**: Using the logit lens distribution at each layer in a set of layers $L$, we construct a candidate set $C$ consisting of the top-$k$ highest probability tokens from each layer: $C_{\text{Head-Projection}} = \bigcup_{\ell \in L} \text{top-}k(D^{(\ell)})$, where $D^{(\ell)} \in \mathbb{R}^{|V|}$ is the logit lens probability distribution

over vocabulary $V$ from layer $\ell$ and top-$k(\cdot)$ returns the highest-probability $k$ elements from each distribution. **Note we limit our experiments to datapoints with single-token answers for simplicity**. We select the top-$k$ tokens for the basic reason that the deleted answer may appear among the top tokens in the logit lens distributions before it disappears from the distributions at later layers (as in Fig. 2). The budget of this attack is $B = k|L|$. For experiments in Sec. 7, we select $k$ and $L$ to optimize attack performance while remaining under a maximum budget $B$ (see tuning details in Appendix A.)

**Probability Delta Attack**: Our second whitebox attack leverages the observation that a "deleted" answer may quickly *rise* and *fall* within the progression of logit lens vocab distributions. We conjecture that by rank-ordering the differences in token probabilities between two consecutive layers, the target answer may be identifiable in the top or bottom $k$ tokens. Consider Fig. 2 again: the deleted answer *Spain* must first rise and later fall significantly across layers as it enters and exits the head of the logit lens distribution. We therefore construct a candidate set as: $C_{\text{Probability-Delta}} = \bigcup_{\ell \in L} \text{top-}k(D^{(\ell+1)} - D^{(\ell)}) \cup \text{bottom-}k(D^{(\ell+1)} - D^{(\ell)})$, where $D^{(\ell)}$ is the logit lens probability distribution from layer $\ell$. This approach constructs a set from the elements that rise and fall the most in the logit lens distributions between layers. For experiments in Sec. 7, we optimize attack performance by tuning $L$ and selecting the top-$k$ elements, bottom-$k$ elements, or union of the two sets (while remaining within a fixed budget of $|C| = 20$). Further tuning details are present in Appendix A.

### 4.2 BLACKBOX ATTACK

**Input Rephrasing**: LLMs are known to be vulnerable to adversarial prompts even after finetuning for chat safety (Zou et al., 2023). For our blackbox attack, we employ a simple but effective technique: prompting with model-generated rephrases of the original input that was used for model editing. Since model editing techniques exhibit good but imperfect generalization across paraphrases (De Cao et al., 2021; Meng et al., 2022), we can extract specific information from a model by rephrasing the input and sampling model outputs across these rephrases (shown in Fig. 2). So, we obtain a candidate set as $C_{\text{Rephrase}} = \bigcup_{r=1}^{R} \{\hat{y}_s \sim P(y|x_r; \mathcal{M}^*)\}_{s=1}^{S}$, where $R$ is the number of rephrases, $S$ is the number of model samples per rephrased input, $x_r$ is the $r$-th rephrasing of $x$ generated by a paraphrasing model, and $P(y|x; \mathcal{M}^*)$ is the output distribution of the edited model given input $x$. We generate $x_r$ using the paraphrasing model from Krishna et al. (2023). The budget of this attack is $|C| = RS$.

## 5 DEFENSE METHODS

**The Empty Response Defense** (Ouyang et al., 2022). This defense employs the basic strategy of optimizing a model to output something *not containing* the sensitive information, which is the strategy behind using RLHF for preventing models from generating sensitive information. We simply optimize the probability of an "empty" target string $d$ with the objective $\arg\max_{\mathcal{M}} p(d|x; \mathcal{M})$, using one of two target strings: "I don't know" and "dummy". The result is that, instead of generating the original knowledge, the model will instead indicate that it does not know the answer ("I don't know" tar-

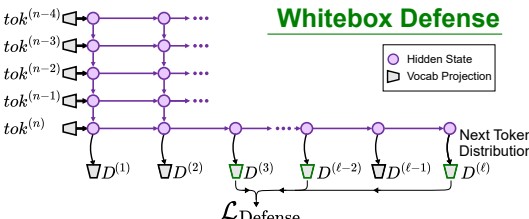

Figure 3: We defend against whitebox attacks by deleting information from intermediate hidden states as well as the final model output distribution (Max-Entropy and Head Projection Defenses).

get) or give some meaningless response ("dummy" target). In our main experiments, we use the "dummy" target, which performs better than using "I don't know" (see Appendix B).

**Fact Erasure** (Hase et al., 2023). Another simple approach to deleting a sensitive answer is to minimize its probability under the model, i.e. minimize $p(y|x; \mathcal{M})$ for the original fact $(x, y)$.

**Error Injection** (De Cao et al., 2021). A common test of model editing methods involves inducing counterfactual knowledge in the model. Here, we use the objective $\arg\max_{\mathcal{M}} p(y^*|x; \mathcal{M})$ where $y^*$ is the alternative, false target provided by Meng et al. (2022). This method would not be applicable in practice, since we do not actually want LLMs to give *wrong* answers to sensitive questions, but we consider it here to show the efficacy of injecting new false information into the model.

**Head Projection Defense**. We introduce a objective that is directly designed to protect against the Head Projection attack. The goal is to prevent the deleted answer from appearing in the top-$k$ elements of the logit lens distributions across a set of layers $L$, as well as the predicted distribution at the final layer (see Fig. 3). To do so, we introduce a max-margin loss in each relevant distribution. With $D^{(\ell)}$ as the logit lens distribution at layer $\ell$, $D^{(\ell)}_{\text{answer}}$ as the original answer's logit lens probability, and $D^{(\ell)}_k$ as the $k$-th top probability in $D^{(\ell)}$, the objective becomes: $\frac{1}{|L|} \sum_{\ell \in L} \max(0, D^{(\ell)}_{\text{answer}} - D^{(\ell)}_k + m)$, where $m$ is the margin term. Since we do not face any constraint over the set of layers $L$ to remove the answer from, we tune over possible layer sets to improve the defense performance (details in Appendix A).

**Max-Entropy Defense**. This defense is similar to the Head Projection Defense, but it varies in terms of the objective for each layer. Here, we maximize the entropy of the model's logit lens distributions over the next token at each layer: $\arg\max_{\mathcal{M}^*} -\frac{1}{|L|} \sum_{\ell \in L} \sum_{y \in |V|} D^{(\ell)}_y \log D^{(\ell)}_y$, where $D^{(\ell)}_y$ is the probability of token $y$ in the logit lens distribution of model $\mathcal{M}^*$ given the input prompt $x$.

**Input Rephrasing Defense**. This defense strategy aims to counter the Input Rephrasing blackbox attack described in Sec. 4. In addition to using the input $x$ for optimization, this approach adds model-generated paraphrases of $x$ to the model editing objective. The rephrased inputs are created using the same off-the-shelf generation model as in the Input Rephrasing attack (Krishna et al., 2023). In other words, for the $i$-th datapoint, we concurrently delete the information for all prompts $x \in x_i \cup X^p_i$, where $X^p_i$ represents the set of rephrases of $x_i$ used for defense. We specifically optimize the Fact Erasure objective in parallel for each input $x \in x_i \cup X^p_i$.

## 6 EXPERIMENT SETUP

**Models**. We conduct experiments with GPT-J (Wang & Komatsuzaki, 2021), Llama-2 (Touvron et al., 2023), and GPT2-XL (Radford et al., 2019). These models were chosen due to their (1) widespread usage, (2) public availability, and (3) capacity for memorizing their pretraining data (Carlini et al., 2023). Results for Llama-2 and GPT2-XL are in Appendix B.

**Datasets**. We use two datasets, CounterFact (Meng et al., 2022) and zsRE (Levy et al., 2017). CounterFact consists of prompts with factual completions, as well as neighboring datapoints that we use for computing Neighborhood $\Delta$-Acc. The zsRE dataset contains short question-answer pairs derived from Wikipedia. Both datasets include alternative, false targets for each input for model editing. To obtain data for computing Random $\Delta$-Acc, after each individual model edit we randomly sample 100 other data points from the respective dataset. **We filter the data to facts that are known by the model we attack**, because it only makes sense to delete facts that are already known by the model. We consider a fact known by the model when the answer string is in the model generation given the prompt; GPT-J gets 34% accuracy on CounterFact data with single-token answers and 25% on zsRE (we also filter to points with single-token answers). Drawing from these eligible facts, our final sample sizes are 587 and 454 datapoints for CounterFact and zsRE respectively when using GPT-J.

**Model Editing Methods**. We employ two popular model editing techniques in our experiments, ROME (Meng et al., 2022) and MEMIT (Meng et al., 2023). We refer the reader to these works for full details of the methods. Both methods work by updating a specific weight matrix in the MLP layer(s) of a Transformer model. When applied to change a single fact in the model, the difference between them is that ROME updates a single layer's MLP (layer 6 for GPT-J by default), while MEMIT updates multiple layers' MLPs (we use layers 5-7). See Appendix A for other method hyperparameters. In Appendix B, we conduct experiments with an additional editing method, constrained finetuning (Zhu et al., 2020), but this method does not perform as well as ROME and MEMIT. Each of our defense methods in Sec. 5 is characterized by its objective function and can be combined with different model editing (optimization) approaches.

## 7 EXPERIMENT RESULTS

### 7.1 CAN WE EXTRACT A "DELETED" ANSWER FROM A LANGUAGE MODEL?

We first ask whether we can attack a model edited with the conventional Empty Response method.

**Design.** We measure Attack-Success@$B$ as a function of the budget $B$ for our three attack methods: the (1) Head Projection, (2) Probability Delta, and (3) Input Rephrasing attacks. We perform these

experiments with GPT-J and CounterFact data, applying the ROME editing method using the Empty Response objective. **We confirm that the edit methods work as designed**: the Rewrite Score is high for all methods (90%+), with low Random $\Delta$-Acc scores (<1%). In general, we increase the budget for whitebox attacks by increasing $|L|$ and $k$, and for our blackbox attack we increase the number of attack paraphrases and the number of samples. See Appendix A for exact hyperparameters.

**Results.** From the results in Fig. 4, we see that attack success reaches values as high as 38% with a budget of $B = 20$. This extremely high attack success rate means that, under our threat model in Sec. 3, the edited model is highly vulnerable to extraction attacks for the "deleted" fact.

Besides the highest attack success rate obtained with whitebox attacks and a budget of $B = 20$, we note that the blackbox attack also achieves a high success rate of up to 29%. Additionally, even with a budget of $B = 1$, an attacker would succeed 18% of the time using the Probability Delta attack. Interestingly, all methods appear to saturate in performance after $B = 20$ candidates.

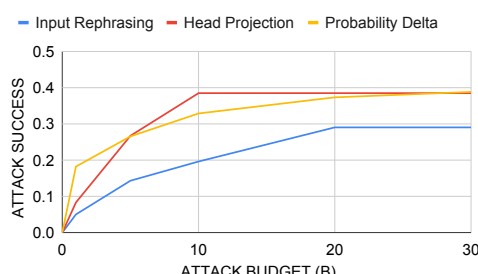

Figure 4: Attack Success vs. the budget $B$ for our three attack methods. We "delete" facts from GPT-J with ROME using the conventional Empty Response objective.

## 7.2 How to Defend Against Information Extraction Attacks

Next, we show how the proposed defense methods (Sec. 5) fare against our extraction attacks (Sec. 4).

**Design.** We evaluate the three baseline methods (Fact Erasure, Empty Resp, and Error Inj) and our three proposed methods (HP Def, Max-Ent Def, and IR Def). We report Attack-Success@$B$ with $B = 20$ using each of our three attack methods, as well as Random $\Delta$-Acc and Neighborhood $\Delta$-Acc metrics to show how much damage is being done to the model's overall knowledge by the deletion of individual facts. We show results for GPT-J on CounterFact and zsRE with ROME and MEMIT.

**Results.** We show the results in Table 1. We summarize the main conclusions as follows:

1. Overall, our whitebox and blackbox attacks are all frequently successful at extracting "deleted" facts. We emphasize that **MEMIT with the Empty Response defense is successfully attacked by our Head Projection attack 89% of the time** on zsRE with $B = 20$. MEMIT is a popular editing method, and the Empty Response objective is the standard approach to preventing models from generating sensitive information, yet this setting totally fails to delete facts from GPT-J.

2. Our Head Projection and Max-Entropy defenses are the strongest defenses against whitebox attacks. Relative to the strongest baseline, Fact Erasure, the Max-Entropy defense lowers the Head Projection attack success by 20.5 points on CounterFact ($22.2\% \rightarrow 1.7\%$ with ROME) and 39.5 points on zsRE ($41.4\% \rightarrow 2.9\%$ with ROME). The Head Projection defense, though helpful, is surprisingly not as effective as the Max-Entropy defense, except for when used with MEMIT on CounterFact. Note we discuss results for the Probability Delta attack below in Sec. 7.3.

3. The Input Rephrasing defense does *not* reduce the blackbox attack success. Across datasets and editing methods, the IR Def never outperforms baselines and is sometimes the worst objective against the Input Rephrasing attack. We confirm that the defense does work when the attack uses the *exact same* paraphrased inputs supplied to the defense's editing objective; the defense fails when attacking paraphrases differ at all. Additionally, we can lower the Input Rephrasing attack success by making the model edits more aggressive, but this has the consequence of skyrocketing $\Delta$-Acc numbers (4.7 for Random data and 27.8 for Neighborhood data; see Appendix B).

## 7.3 Can We Defend Against Unforeseen Extraction Attacks?

Lastly, we examine the efficacy of extraction attacks that the defense methods are not directly designed to prevent, an important "unforeseen" scenario for our defense methods (Carlini et al., 2019).

**Design.** We highlight results from our previous experiment, specifically the performance of the Probability Delta Attack applied against our two whitebox defenses, Max-Entropy and Head Projection.

| Defense | Attack-Success@20 | | | Δ-Acc | | Rewrite Score |
|---------|-------------------|--|--|-------|--|--------------|
| | Head Projection | Probability Delta | Input Rephrasing | Random | Neighbors | |
| CounterFact | | | | | | |
| ROME | | | | | | |
| + Fact Erasure | 22.15 | 25.38 | 22.83 | 0.72 | 8.74 | 99.69 |
| + Empty Resp | 36.84 | 37.65 | 29.02 | 0.54 | 3.76 | 99.58 |
| + Error Inj | 99.20 | 99.83 | **20.10** | 1.00 | 9.60 | 99.30 |
| + HP Def | 4.43 | 20.27 | 27.77 | 0.69 | 6.35 | 99.73 |
| + Max-Ent Def | **1.70** | **2.39** | 27.94 | 0.69 | 6.27 | 99.73 |
| + IR Def | 56.39 | 60.65 | 29.30 | 0.69 | 6.17 | 98.88 |
| MEMIT | | | | | | |
| + Fact Erasure | 39.18 | 46.17 | **34.07** | 0.26 | 3.29 | 98.68 |
| + Empty Resp | 67.09 | 72.60 | 49.31 | 0.22 | 1.03 | 87.54 |
| + Error Inj | 98.60 | 98.80 | 36.38 | 0.15 | 2.05 | 97.68 |
| + HP Def | **19.42** | **38.33** | 42.76 | 0.20 | 3.37 | 97.09 |
| + Max-Ent Def | 34.24 | 39.01 | 50.77 | 0.19 | 3.32 | 96.41 |
| + IR Def | 56.03 | 61.50 | 41.91 | 0.20 | 3.49 | 91.56 |
| zsRE | | | | | | |
| ROME | | | | | | |
| + Fact Erasure | 41.41 | 43.83 | 15.64 | 0.10 | - | 94.80 |
| + Empty Resp | 36.83 | 59.13 | 13.00 | 0.08 | - | 99.78 |
| + Error Inj | 20.68 | 45.07 | **10.35** | 0.13 | - | 99.40 |
| + HP Def | 31.28 | 56.83 | 18.50 | 0.12 | - | 90.53 |
| + Max-Ent Def | **2.86** | **2.42** | 18.50 | 0.12 | - | 90.71 |
| + IR Def | 84.80 | 84.80 | 29.07 | 0.07 | - | 80.64 |
| MEMIT | | | | | | |
| + Fact Erasure | 42.51 | 42.07 | **22.18** | 0.05 | - | 91.34 |
| + Empty Resp | 88.55 | 88.11 | 29.30 | 0.05 | - | 91.34 |
| + Error Inj | 89.65 | 80.64 | 32.60 | 0.11 | - | 85.86 |
| + HP Def | 53.52 | 72.47 | 28.19 | 0.05 | - | 89.46 |
| + Max-Ent Def | **39.92** | **38.11** | 29.74 | 0.07 | - | 91.24 |
| + IR Def | 46.04 | 57.93 | 27.75 | 0.07 | - | 84.16 |

Table 1: Attack success rates of the the three proposed attacks (Sec. 4) across defense methods (Sec. 5), for facts from CounterFact and zsRE that are known by GPT-J.

In this setting, we use defenses that were designed to protect against the Head Projection attack but *were not designed to defend against our second whitebox attack*, the Probability Delta attack.

**Results.** We draw a few conclusions from Table 1: (1) The "unforeseen" Probability Delta attack is very effective against the Head Projection defense, which was not prepared for it. (2) Our Max-Entropy defense often helps against the Probability Delta attack despite not being specially designed for it. Compared to the Head Projection defense on zsRE, Max-Entropy defense substantially lowers attack success rates (56.8%→2.4% with ROME and 72.5%→38.1% with MEMIT). However, while the Max-Entropy defense can lower whitebox attack success to 2.4%, (3) blackbox attack success remains quite high at 28% for CounterFact and 19% for zsRE, suggesting that the defense is still inadequate against blackbox attacks. In total, we see that there is no single defense method that is prepared against all attacks it could face, even if it is effective against some unforeseen attacks.

# 8 CONCLUSION

We first argue that model editing methods are the most promising approach to deleting sensitive information from LLMs, rather than interventions focusing on pretraining and finetuning data. Even for this promising class of methods, however, we show that "deleted" information can be extracted a surprisingly high percentage of the time (as high as 89% in some experiments) when the attacker operates with a small budget of verification attempts $B$. We motivate this budget via a threat model based on three plausible adversarial settings. Our findings suggest that truly deleting sensitive inormation is a tractable but difficult problem, with potentially severe implications for deployment of LLMs in a world where individuals enjoy a robust right to privacy and safety from harmful model outputs.

## ETHICS STATEMENT

This paper addresses problems involving sensitive information in large language models. This is an important topic with serious ethical implications, as language models currently possess knowledge that could be dangerous to humans and output directly harmful text. We hope that the technical methods in this paper can help mitigate these important ethical problems, but at the same time, we want to demonstrate that it may be fundamentally difficult to solve the problem of sensitive information in pretrained language models. These results could imply that there are negative moral and legal consequences to deploying LLMs in situations where they may influence humans. We leave it for future work in AI, ethics, and law to fully explore the implications of work on sensitive information deletion and LLMs.

## ACKNOWLEDGEMENTS

We thank Neel Nanda for helpful experiment suggestions. This work was supported by NSF-CAREER Award 1846185, NSF-AI Engage Institute DRL-2112635, DARPA MCS Grant N66001-19-2-4031, and Google PhD fellowship. The views contained in this article are those of the authors and not of the funding agency.

## REPRODUCIBILITY STATEMENT

We have provided the code for all experimental results in the supplementary materials to facilitate reproducibility. Additionally, comprehensive hyperparameters and other essential details required for replication can be found in Appendices A and D.

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

## A  TUNING DETAILS

For a fixed budget of $B = 20$, we tune over the hyperparameters that control the budget on a separate development set of 100 samples.

**Whitebox Attacks:**  Given a fixed candidate set size represented by $B$, we seek to optimize the allocation of resources by tuning two parameters: $k$ and $L$. These parameters determine the distribution of the available budget across $L$ layers, while retaining the $k$ candidate tokens from each layer in the set $C$ such that $kL = B$.

We try different combinations of $k$ and $\ell$ in the following set [(1, 20), (2, 10), (4, 5), (5, 4), (2, 10), (1, 20)] for each of the options when choosing k candidates (top-$k$, bottom-$k$ and (top-$k/2 \cup$ bottom-$k/2$)). For each $k$, we choose the set $L$ such that $|L| = \ell$ and Attack-Success@$B$(k, L) is maximum when evaluated on the development set.

For the Head-Projection Attack, we find that choosing top-4 ($k = 4$) highest probability candidates from each layer's intermediate distribution while keeping 5 optimal layers (17, 18, 19 20, 21)

($|L| = 5$) tuned on a separate development set of 100 sample points attains the highest attack success rate for GPT-J.

For the Probability Delta attack, we find that choosing top-2 highest probability candidates (rather than both top-k and bottom-k) from each layer's intermediate distribution while keeping 10 ($|L| = 10$) layers 8-9, 16-23 is optimal for the attack success for GPT-J, 36-45 for GPT2-XL.

| Model | Attack-layers-HP | Attack-layers-PD | Defense-layers |
|---|---|---|---|
| GPT-J-6B | 17-21 | 8-9, 16-23 | 8-9, 16-28 |
| GPT2-XL-1.5B | 41-45 | 36-45 | 36-48 |
| Llama2-7B | 23-27 | 21-30 | 23-32 |

Table 2: Hyperparameters following tuning of attack and defense methods.

**Blackbox Attacks:** Given a fixed candidate set size represented by $B$, we seek to optimize the allocation of resources by tuning two parameters: $R$ and $S$. These parameters determine the distribution of the available budget $R$ model-generated paraphrases, while randomly sampling the $s$ candidate tokens from each paraphrase in the set $C$ such that $RS = B$. We try different combinations of $n$ and $s$ in the following set [(1, 20), (2, 10), (4, 5), (5, 4), (2, 10), (1, 20)]. For the Input Rephrasing attack, we find that using 4 ($R = 4$) paraphrases with 5 ($S = 5$) samples from each paraphrase in the candidate set leads to highest attack success rate on the development set.

**Whitebox Defense:** For the HP and PD defenses, we choose a set of layers $L$ consisting of the attack layers as well as all the subsequent layers starting from the last attack layer to the final layer so as to propagate the defense to the final layers. See the final selected layers in Table 2.

**Blackbox Defense:** We choose 5 paraphrases for this defense which are obtained using the model (See Appendix D for the model details and hyperparameters).

# B ADDITIONAL EXPERIMENTS

## B.1 LLAMA2-7B

We show results for Llama2-7b in Table 3. We modify some of the default edit parameters for of ROME so as to make the edits suitable for Llama-2-7b (Touvron et al., 2023) i.e. a reasonable rewrite score and delta accuracy. These are the hyperparameters used in ROME that we modify and their values for reproducibility: Learning rate (v_lr): 5e-1, Loss layer (v_loss_layer): 31, Weight decay factor (v_weight_decay): 5e-5, The threshold at which the norm of the change vector (the norm of difference between original and new vector $v$ is clamped at) (clamp_norm_factor): 2.

| Method | Defense | Attack-Success@20 | | | $\Delta$-Acc | | |
|---|---|---|---|---|---|---|---|
| | | Head Projection | Probabaility Delta | Input Rephrasing | Random | Neighborhood | Rewrite Score |
| | | | CounterFact | | | | |
| ROME | Fact-erasure | 15.93 | 25.95 | 51.43 | 3.30 | 16.20 | 98.53 |
| | Empty Resp | 62.14 | 62.86 | 56.14 | 3.09 | 15.23 | 74.59 |
| | Error Inj | 64.86 | 63.14 | 56.29 | 2.86 | 15.50 | 75.11 |
| | HP Def | 3.43 | 4.71 | 55.14 | 3.38 | 16.77 | 96.22 |
| | Max Ent def | 3.14 | 4.57 | 52.86 | 3.38 | 17.06 | 99.27 |
| | IR def | 65.57 | 65.43 | 70.14 | 0.69 | 4.70 | 68.24 |

Table 3: Attack success rates of the the three proposed attacks (Sec. 4) on the CounterFact dataset when information is deleted from the Llama2-7B model using ROME augmented with the defense strategies (Sec. 5)

## B.2 GPT2-XL

We report GPT2-XL-1.5B numbers in Table 4, and we show attack success as a function of attack budget in Fig. 5.

| Defense | Attack-Success@20 | | | Δ-Acc | | Rewrite Score |
|---|---|---|---|---|---|---|
| | Head Projection | Probability Delta | Input Rephrasing | Random | Neighbors | |
| CounterFact | | | | | | |
| GPT2-XL, ROME | | | | | | |
| Fact-erasure | 35.57 | 40.86 | 6.14 | 1.14 | 3.09 | 97.32 |
| HT-def | 6.29 | 41.43 | 5.57 | 1.17 | 3.09 | 97.34 |
| Max-Ent Def | 6.29 | 41.43 | 4.71 | 1.18 | 3.09 | 97.34 |
| Empty resp | 28.14 | 34.29 | 5.57 | 1.31 | 1.46 | 99.73 |
| IR def | 34.57 | 72.14 | 27.43 | 0.98 | 1.59 | 98.78 |
| Err Inj | 99.72 | 99.99 | 3.4 | 1.5 | 3.1 | 98.03 |
| GPT2-XL, MEMIT | | | | | | |
| Fact-erasure | 55.57 | 41.71 | 17.71 | 0.57 | 1.53 | 98.16 |
| Empty resp | 87.00 | 89.71 | 50.43 | 0.38 | 0.54 | 99.95 |
| Err Inj | 89.32 | 89.72 | 31.72 | 0.53 | 1.39 | 99.54 |
| HP-def | 21.00 | 41.29 | 18.00 | 0.68 | 3.17 | 97.50 |
| Max-Ent Def | 21.86 | 61.29 | 18.14 | 0.67 | 3.13 | 97.55 |
| IR def | 61.43 | 64.86 | 32.29 | 0.91 | 2.90 | 97.59 |
| GPT2-XL, FT | | | | | | |
| Fact-erasure | 93.43 | 96.14 | 50.43 | 2.49 | 3.26 | 96.46 |
| Err Inj | 98.70 | 24.30 | 60.08 | 1.60 | 0.8 | 99.99 |
| Empty resp | 99.57 | 99.86 | 59.14 | 2.49 | 1.51 | 99.73 |
| IR def | 94.00 | 96.29 | 52.71 | 2.47 | 2.63 | 97.02 |

Table 4: Attack success rates of the the three proposed attacks (Sec. 4) on the CounterFact dataset when information is deleted from the GPT2-XL model using the three editing methods augmented with the defense strategies (Sec. 5)

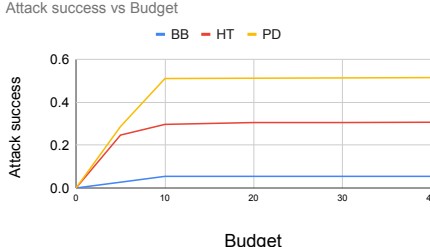

Figure 5: As the attack budget increases, the attack success increases and saturates after a budget of 10. Here budget for HP and PD attacks is 20 and that for BB (IR) attack is 10. Here the editing method is Fact erasure and model is GPT2-XL.

## B.3 CONSTRAINED FINETUNING

**Constrained Finetuning** (Zhu et al., 2020). Here, we employ a simple optimization approach based on the Adam optimizer, incorporating an $\ell_\infty$-norm constraint, as outlined by (Zhu et al., 2020). We finetune the same singular MLP weight matrix that we perform edits to in ROME.

| Method | Defense | Attack-Success@20 | | | Δ-Acc | | Rewrite Score |
|---|---|---|---|---|---|---|---|
| | | Head Projection | Probabaility Delta | Input Rephrasing | Random | Neighborhood | |
| | CounterFact | | | | | | |
| FT | Fact-erasure | 47.53 | 93.02 | 29.09 | 4.27 | 27.02 | 95.89 |
| | Empty resp | 78.94 | 80.70 | 61.69 | 0.70 | 5.57 | 97.51 |
| | Error Inj | 54.43 | 60.94 | 56.29 | 2.86 | 15.50 | 75.11 |
| | HP-def | 99.15 | 96.76 | 75.81 | 0.02 | 0.05 | 0.17 |
| | Max-Ent-def | 99.32 | 96.93 | 75.81 | 0.00 | 0.00 | 0.00 |
| | IR def | 46.85 | 58.77 | 13.12 | 4.25 | 26.95 | 95.90 |

Table 5: Attack success rates of the the three proposed attacks (Sec. 4) on the CounterFact dataset when information is deleted from the GPT-J model using Constrained Finetuning (FT) augmented with the defense strategies (Sec. 5)

| Defense | Attack-Success@20 | | | Δ-Acc | | Rewrite Score |
|---|---|---|---|---|---|---|
| | Head Projection | Probability Delta | Input Rephrasing | Random | Neighbors | |
| CounterFact | | | | | | |
| ROME | | | | | | |
| + Fact Erasure | 22.15 | 25.38 | 22.83 | 0.72 | 8.74 | 99.69 |
| + Empty Resp (*dummy*) | 36.84 | 37.65 | 29.02 | 0.54 | 3.76 | 99.58 |
| + Empty Resp (*I don't know*) | 54.89 | 55.92 | 32.80 | 0.50 | 3.20 | 98.74 |

Table 6: Attack success rates of the the three proposed attacks (Sec. 4) across defense methods (Sec. 5), for facts from CounterFact and zsRE that are known by GPT-J.

## C  EDITING METHODS AND ADVERSARY MODEL ACCESS

### C.1  MODEL EDITING METHODS

**ROME** (Meng et al., 2022). Rank-One Model Editing (ROME) is a state-of-the-art method that changes model outputs by updating a specific MLP layer in the model (layer 6 for GPT-J by default). The update is applied to the second matrix within this MLP layer, and the update itself is constrained to be a rank-one matrix that is obtained analytically when treating the MLP weight as a linear associative memory (Meng et al., 2022). The default objective that is maximized by this update is the model probability $p(y^*|x)$ for a desired output $y^*$, with data augmentation for the input $x$ and some regularization. Recall that we can apply this editing method to optimize any of the defense objectives from Sec. 5.

**MEMIT** (Meng et al., 2023). MEMIT is a method designed for updating an arbitrary number of facts in a model, as opposed to a single fact. When applied to update only one fact, however, its only difference from ROME is that it "spreads out" its update over *multiple MLP layers* rather than a single MLP layer as in ROME (Meng et al., 2023). When applying MEMIT to GPT-J, we update layers 5-7.

### C.2  MODEL ACCESS

Our blackbox attack relies on simple random sampling that can be carried out through the OpenAI API as of September 28, 2023: https://platform.openai.com/docs/guides/gpt. Though we perform experiments with publicly available models, this is important since many models of interest are gated behind APIs.

*Parameters for OpenAI API:*

Input Prompts or Instructions: Users send input prompts or instructions to the OpenAI API. These prompts provide context or guidance to the model regarding the task or the type of response expected.

Sampling Parameters: To customize the behavior of the model and the characteristics of the generated text, users can specify various sampling parameters:

- Sampling Temperature (temperature): This parameter controls the randomness of the generated text. Higher values (e.g., 0.8) make the output more random, while lower values (e.g., 0.2) make it more deterministic.

- Maximum Tokens (max_tokens): Users can limit the length of the generated text by setting a maximum number of tokens. This helps in controlling the response length.

- Top-p Probability (top_p): This parameter allows users to set a probability threshold for the next token. Tokens with probabilities above this threshold are considered, which helps in influencing the diversity of generated responses.

- Frequency Penalty (frequency_penalty): Users can penalize the repetition of words in the generated text by adjusting this parameter. Higher values discourage word repetition.

- Presence Penalty (presence_penalty): This parameter allows users to penalize the presence of certain words or phrases in the generated text. It can be useful for controlling the content or style of the output.

- Stop Sequences (stop_sequences): Users can specify strings that the model should avoid generating in the output. This is helpful for preventing specific content from appearing in the generated text.

- Temperature Schedule (temperature_schedule): Users can provide a list of temperature values to change the temperature dynamically during the text generation process. This can result in text that starts more deterministic and becomes more random over time, or vice versa.

- Top-k Tokens (top_k): This parameter limits the number of tokens considered at each step of generation to the top-k most likely tokens. It can help in controlling the model's creativity and focus.

API Response: After sending the input prompt and specifying the desired sampling parameters, users receive the model-generated text or response from the OpenAI API. The output text is influenced by the provided context and the parameter settings.

## D    REPRODUCIBILITY DETAILS

Here, we give additional details to our experiments that would be necessary for reproducing the results.

**Paraphrase Model.**    We use the dipper-paraphraser-xxl (Krishna et al., 2023) model available on huggingface. We first generate paraphrases of the entire prompt, including the target answer by varying the following parameters in the model: lexical diversity in [20, 40, 60, 80], order diversity in [20, 40, 60, 80], top_p in [0.25, 0.5, 0.75]. We then retain only the paraphrases which have the target answer as the last word and obtain the paraphrased prompt by truncating the paraphrased sentence to remove the last word which is the target answer.

**$\Delta$-Acc Metrics.**    The length of generated output that we use for measuring $\Delta$-Acc is 36.

**Rewrite Score.**    We consider the Rewrite Score from Hase et al. (2023) as a traditional measure of edit success, to be reported alongside Attack-Success metrics. The Rewrite Score measures how much the edit changes the new target probability as a fraction of the possible desired change:

$$\frac{p(y|x; \mathcal{M}^*) - p(y|x; \mathcal{M})}{1 - p(y|x; \mathcal{M})}$$

A value of 1 means that the edit perfectly maximizes the new target probability, while a value of 0 means that the new target probability did not change at all. When the probability of a target is being minimized rather than maximized (which occurs in some defense objectives), this metric simply becomes $1 - p(y|x; \mathcal{M}^*)/p(y|x; \mathcal{M})$, reflecting that we desire the target probability to approach 0. Specifically, we use the original formulation for a maximizing objective with Empty Response and Error Injection, and we use the simplified version $(1 - p(y|x; \mathcal{M}^*)/p(y|x; \mathcal{M}))$ when reporting Rewrite Score for Fact Erasure, Head Projection, Probability Delta, and Input Rephrasing defenses, since these methods involve *lowering* the probability of the target answer.

**Head Projection defense.**    We backpropagate though both $D_{\text{answer}}^{(\ell)}$ and $D_k^{(\ell)}$ without the use of any stop-gradient. (See Sec 5).

**Data Filtering.**    On top of the single-token filtering, we also require the original model probability $p(y|x; \mathcal{M})$ of the correct target answer which is being deleted to be at least 0.02, in order for it be meaningful to measure a decrease in the next-token probability.

## E    RESULTS DISCUSSION

Attack success against MEMIT is generally higher than against ROME, and our best defense methods (like Max-Ent) achieve smaller improvements over baselines with MEMIT than with ROME. This could suggest that distributing editing updates across multiple layers rather than a single layer, as

MEMIT does, increases vulnerability to attacks and makes defense more challenging. However, MEMIT performs more favorably than ROME on the Rewrite Score and $\Delta$-Acc metrics, meaning this difference in the methods could be due to a difference in how strongly the edits applied to the model.

