# OpenReview forum: "Can Sensitive Information Be Deleted From LLMs? Objectives for Defending Against Extraction Attacks"
_ICLR.cc/2024/Conference — ICLR 2024 spotlight_

### Official Review · Reviewer_KWLr · 2023-10-31

**Soundness:** 4 excellent
**Presentation:** 3 good
**Contribution:** 3 good
**Rating:** 8
**Confidence:** 4

**Summary:**

The paper shows that models edited by model editing methods ROME and MEMIT still contain sensitive information and that the information is not fully "deleted" from the model. Several white-box and one black-box attack are proposed which are used to extract information from edited models, given that the attacker has an attack budget B. If the sensitive information is in within the B output candidates the information is assumed to be leaked. In addition to the attacks, the paper proposes multiple defense methods. Both the attacks and defenses are evaluated on the Counterfact and zsRE dataset. The evalution shows that the defense methods are not enough to defend against extraction attacks and that even in a black-box setting, information can still be extracted after editing the model.

**Strengths:**

- the paper is well written and easy to follow
- code and everything to reproduce the experiments is given
- the evaluation is quite thorough, evaluating multiple defenses against multiple attacks

**Weaknesses:**

For me it is not quite clear how the proposed defense methods are combined with the model editing techniques. In the experiments, ROME and MEMIT are used as model editing techniques. However, it is not mentioned how the different optimization objectives for the proposed defense techniques are used in combination with these methods. This could be formulated a bit clearer.

Misc:
- it would be easier to read if the paragraph in 4.2 also had a bold subheading with the name of the attack, instead of putting the name in the heading of the section. This would make it easier for readers to spot the names of the different attacks.

**Questions:**

- **Q1:** Why use only single-token answers? If I understand this correctly, this way it is not possible to extract answers which consist of multiple tokens, correct? Is it possible to modify your approach to make this work for multi-token answers?
- **Q2:** I don't quite understand how the Head Projection defense works. What exactly are the values D_answer and D_k? As far as I understand, D_answer is a probability distribution, while D_k is a single value? Could you clarify the loss function and what exactly is optimized for this defense?

**Details Of Ethics Concerns:**

All ethical considerations are addressed in the ethics statement, which is why no ethical review is needed.

---

> ### Author Response · Authors · 2023-11-17
> **Reply to Reviewer KWLr**
>
> > It is not mentioned how the different optimization objectives for the proposed defense techniques are used in combination with these methods.
>
> We apologize for confusion about this. To clarify, the defense methods are objectives, and the underlying optimization method is ROME/MEMIT i.e. the module in the model which is being edited is decided by the optimization method and the objective for which the edit is optimized is decided by the defense method. So in Table 1, we list ROME and MEMIT as the optimization methods, and the +X is the defense objective that is optimized by the optimization method.
>
> We will be able to make this clearer with extra space in a camera ready version of the paper.
>
> > paragraph in 4.2 also had a bold subheading
>
> Thanks. We’ll make it bold in the final version.
>
>
> > Is it possible to modify your approach to make this work for multi-token answers?
>
>
>
> On the attack side, yes it would be possible to extend our attacks to the multi-token setting. For the whitebox attacks, the logit lens could be readily applied in a manner similar to autoregressive decoding to generate outputs or rank-order a set of plausible options (i.e. doing autoregressive decoding but decoding from an intermediate layer’s hidden states rather than running the full forward pass). On the blackbox side, generating multiple tokens is straightforward.
>
> We note that the main reason for this simplifying assumption is that it makes it easier to rank order and evaluate the attack success. If we’re deciding whether the ground truth answer is in the candidate set, it's easier if there are 50000 possible atomic answers to questions, and we can directly check whether the exact answer is in the candidate set. If the space of answers is really big, that basically turns into a complicated NLG evaluation problem (is the true answer similar enough to some candidate in the candidate set?), so we wanted to simplify the problem in our work.
>
>
> > I don't quite understand how the Head Projection defense works. What exactly are the values D_answer and D_k? As far as I understand, D_answer is a probability distribution, while D_k is a single value? Could you clarify the loss function and what exactly is optimized for this defense?
>
> The objective of HP defense is to prevent the deleted answer from appearing in the top-k elements of the logit lens distributions across a set of layers denoted as L. This is done to defend against the HP attack (refer to Fig. 3). To achieve this goal, we introduce a max-margin loss in each relevant distribution. Here, D(ℓ) represents the logit lens distribution at layer ℓ, D(ℓ)_answer denotes the probability of the original answer's logit lens, and D(ℓ)_k represents the k-th top probability in D(ℓ). The given margin loss objective forces the probability of answer (D(ℓ)_answer) to be less than the probability of the kth highest-probabiity token (D(ℓ)_k) by at least a margin of m which makes the answer less detectable in the HP attack (see Sec 5, page number 7).

---

> > ### Comment · Reviewer_KWLr · 2023-11-22
> > **Thanks for the clarification**
> >
> > Thank you for the clarification. My questions have been appropriately addressed, which is why I am maintaining my score of "accept".

---

### Official Review · Reviewer_X7HP · 2023-10-31

**Soundness:** 3 good
**Presentation:** 3 good
**Contribution:** 2 fair
**Rating:** 6
**Confidence:** 3

**Summary:**

This paper studies whether a piece of information can be effectively deleted from an LLM. The authors adapt two model editing techniques to the task of suppressing a piece of information from an LLM. They then evaluate the robustness of these techniques, combined with different defense approaches, to several new attacks (both black-box and white-box) aiming to extract that piece of information from the LLM. The authors propose a new threat model, where the attack is considered successful if the targeted information is among B candidates produced by the attack. Empirical results using one of the model editing techniques, ROME, show that it’s not always effective, as the targeted information can still be extracted 38% of the time in a white-box setting and 29% of the time in a black-box setting. The authors further show that defense techniques can reduce the risk at a small cost in utility, but that in some cases they still remain vulnerable to attacks they don’t explicitly protect against.

**Strengths:**

- 1) Important problem: can information be deleted from an LLM? The premise of the paper, that the right way to delete information is to modify the model post-hoc instead of curating the training dataset, is quite contentious. In spite of this, for practical reasons model developers might indeed not curate their training data, which motivates the need to evaluate the robustness of model editing techniques.
- 2) Well-motivated threat model, based on the insight that considering some information to be deleted only if it cannot be recovered directly (B=1) is an insufficient requirement.
- 3) The technical contribution of the paper (attacks and defenses) is insightful, well motivated and clearly described.

**Weaknesses:**

- 1) The findings of the paper are somewhat expected, as the model editing techniques being evaluated are heuristics and don’t come with formal guarantees of robustness against attacks. Similarly, it is expected that a defense designed to mitigate a specific attack is robust against that attack but not necessarily against other attacks.
- 2) Insufficient analysis of results. I was left wondering what are the technical differences between ROME and MEMIT and whether this could explain some of the differences in the results.

Minor (suggestions for improvement):
- 3) Confusing usage of the term “sensitive”: The definition used by the authors includes “toxic” information: “Models can also generate text reflecting beliefs that cause direct psychological harm to people (i.e. toxic generated text) (Kenton et al., 2021). Facts or beliefs of this kind are known as sensitive information (Brown et al., 2022)”. I'm pretty sure that’s not how Brown et al. use the word sensitive. In the privacy domain, “sensitive information” refers to protected characteristics about an individual (https://commission.europa.eu/law/law-topic/data-protection/reform/rules-business-and-organisations/legal-grounds-processing-data/sensitive-data/what-personal-data-considered-sensitive_en) or is sometimes used colloquially to denote private information that a model or system should not disclose. To avoid confusion, I would suggest using a different term that explicitly refers to “toxic” information.

**Questions:**

- 1) What are the results of attacking the MEMIT method without any defenses (i.e., the equivalent of Figure 4)? The paper’s second claim is that model editing methods fail to delete information; basing it on only one of two editing methods studied in the paper weakens the claim and raises questions.
- 2) Do the authors think that model editing is technically possible with formal guarantees against attacks and what are, in the authors’ opinion, promising directions for future work in this domain?

---

> ### Author Response · Authors · 2023-11-17
> **Reply to Reviewer X7HP**
>
> > The findings of the paper are somewhat expected
>
> True, they are expected but they are also important and necessary since there has been a lot of excitement around editing methods as well as many concerns about privacy and memorized information in language models. This is a timely policy issue in society, and it’s not clear that we have the technical tools to address the problem at the moment.
>
> At the same time, one surprising result is that we do see that HP Def works pretty well against PD attack, which it is not prepared for. In the paper, we suggest that there is no single defense method which works against all attacks it could face, but we do see some surprisingly positive results within the whitebox attack category.
>
> Additionally, some readers will probably be surprised to see that “deleted” information is still recoverable from model hidden states in the first place. This has not been shown before empirically.
>
> > I was left wondering what are the technical differences between ROME and MEMIT and whether this could explain some of the differences in the results.
>
> Thanks for pointing this out, we actually did discuss this in an earlier, longer version of the paper but unfortunately our ability to comment on every difference in methods was constrained by limited space in the ICLR submission. We are happy to further discuss these kinds of details (see below) in an extra camera ready page.
>
> Regarding the differences between ROME and MEMIT: Attack success against MEMIT is generally higher than against ROME, and our best defense methods (like Max-Ent) achieve smaller improvements over baselines with MEMIT than with ROME. This could suggest that distributing editing updates across multiple layers rather than a single layer, as MEMIT does, increases vulnerability to attacks and makes defense more challenging. However, MEMIT performs more favorably than ROME on the Rewrite Score and ∆-Acc metrics, meaning this difference in the methods could be due to a difference in how strongly the edits applied to the model.
>
> > Confusing usage of the term “sensitive”
>
> Thanks for pointing this out, we will clarify the different meanings of sensitive vs. toxic (with appropriate citations to broader socio-technical work) in the final version of the paper.
>
> >  The paper’s second claim is that model editing methods fail to delete information; basing it on only one of two editing methods studied in the paper weakens the claim and raises questions.
>
> It is true that Sec. 7.1 focuses on ROME in order to make the point that SOTA model editing methods are vulnerable to information attacks. However, we believe this result would only hold more strongly for the MEMIT and constrained finetuning methods, since we see in results in 7.2 that MEMIT with defenses is more easily attacked compared to ROME for almost all defenses. So we expect the trend to be similar for MEMIT but with even higher attack success rates (see also previous answer on ROME vs MEMIT).
>
> > Do the authors think that model editing is technically possible with formal guarantees against attacks and what are, in the authors’ opinion, promising directions for future work in this domain?
>
> This is a good question! To our understanding of the adversarial robustness literature, formal guarantees often (1) make strong assumptions about the form of the attack, and (2) are often limited to data domains like images rather than text. It seems to us that the problem of information extraction attacks on LMs is still a sufficiently new problem that additional empirical and analysis work needs to be conducted first before a stronger theoretical understanding is achievable, although clearly formal guarantees of robustness are very relevant and valuable. Interestingly, there is also the opinion that as models get better (in both vision and language settings), eventually adversarial vulnerability will go away on its own. For instance, you might think that paraphrase attacks cease to be effective once models truly understand that we don’t want them to reveal people’s private information no matter how they are asked about it. Overall, this is still a fledgling area of research, but since it is very practically important it demands much more attention.

---

### Official Review · Reviewer_GnJt · 2023-11-04

**Soundness:** 4 excellent
**Presentation:** 4 excellent
**Contribution:** 3 good
**Rating:** 8
**Confidence:** 3

**Summary:**

This paper investigates model editing methods to remove sensitive information from LLM. This work shows that "deleted" information can be extracted from the hidden state when the attacker uses a smaller budget of verification attempts. Simple rewriting of prompts can also cause LLM to generate sensitive information. This is an interesting study, and the attack and defense methods it provides are worthy of further study and discussion.

**Strengths:**

This work elaborates on the security issues of LLMs from the perspective that hidden states may leak sensitive information. It presents potential attack methods and defense strategies.

**Weaknesses:**

This work provides an incomplete description of the reasons behind some experimental phenomena. The reasons or intuitions why defense strategies based on data augmentation do not work are not revealed.

**Questions:**

This paper shows some interesting results. My main question is whether the author can give more detailed insights or possible mechanisms for the findings in the paper. For example, why do the hidden states of LLMs reveal sensitive information? What is the intuition behind this phenomenon? Why are defense strategies based on data augmentation ineffective?

In addition, can fine-tuning, a typical defense strategy, be combined with the defense scheme (e.g., Head Projection Defense) proposed in this paper to produce a more powerful defense method?

---

> ### Author Response · Authors · 2023-11-17
> **Reply to Reviewer GnJt**
>
> > why do the hidden states of LLMs reveal sensitive information? What is the intuition behind this phenomenon?
>
> We leverage the insight from interpretability research that output information accrues over time in the hidden states of a Transformer forward pass [1,2,3,5]  (See Sec 4.1)
>
> Other past work shows that models memorize their pretraining data, often a large proportion (like 1%) [4].
>
> As a result, we believe sensitive information is memorized and accrues across layers during the forward pass. What our paper adds to this perspective, is that we show that edits to the model do not properly remove information across all the layers, and it can still be found by examining several layers' latent states.
>
> > This work provides an incomplete description of the reasons behind some experimental phenomena…Why are defense strategies based on data augmentation ineffective?
>
> It is true that the Input Rephrasing defense that uses data augmentation to delete information for multiple paraphrases of the input data does not work against blackbox attacks. We confirm that the defense does work when the attack uses the *exact same* paraphrased inputs supplied to the defense’s editing objective; the defense fails when attacking paraphrases differ at all, even if they are drawn from the same *distribution*. Additionally, we can lower the Input Rephrasing Attack success by making the model edits more aggressive, but this has the consequence of skyrocketing ∆-Acc numbers. (See Sec 7.2)
>
> > can fine-tuning, a typical defense strategy, be combined with the defense scheme (e.g., Head Projection Defense) proposed in this paper to produce a more powerful defense method?
>
> Yes! Finetuning is an optimization method and is agnostic to the objective. Each of our defense methods is characterized by its objective function and can be combined with different model editing (optimization) approaches, which are described in Sec. 6. See Appendix B.3 and Table 5 in appendix where we combine our defense objectives with constrained fine-tuning optimization method. With an extra camera ready page we will flag this point more prominently than in the original draft.
>
> [1] nostalgebraist. interpreting gpt: the logit lens, 2020.URL https://www.lesswrong.com/ posts/AcKRB8wDpdaN6v6ru/interpreting-gpt-the-logit-lens
>
> [2] Belrose, Nora, et al. "Eliciting latent predictions from transformers with the tuned lens." arXiv preprint arXiv:2303.08112 (2023).
>
> [3] Pal, Koyena, et al. "Future Lens: Anticipating Subsequent Tokens from a Single Hidden State." arXiv preprint arXiv:2311.04897 (2023).
>
> [4] Carlini, Nicholas, et al. "Quantifying Memorization Across Neural Language Models." The Eleventh International Conference on Learning Representations. 2022.
>
> [5] Geva, Mor, et al. "Transformer Feed-Forward Layers Are Key-Value Memories." Proceedings of the 2021 Conference on Empirical Methods in Natural Language Processing. 2021.

---

### Official Review · Reviewer_174w · 2023-11-06

**Soundness:** 2 fair
**Presentation:** 3 good
**Contribution:** 3 good
**Rating:** 8
**Confidence:** 3

**Summary:**

This paper investigates the removal of memorized information from language models via direct editing of model weights. The paper first establishes a threat model for  LLM extraction, based on the notion of recovery of sensitive information from $B$ candidates. The paper then describes several new attacks: two variants of whitebox attacks, in which an attacker may utilize probabilities computed from intermediate hidden states in the model to aid in recovery, and a blackbox attack, in which an input query is “rephrased'' multiple times by the model to generate a diverse set of candidate responses. Finally, the paper describes and evaluates several defenses against the attacks, which are able to be applied in conjunction with existing model editing techniques.

**Strengths:**

- Paper works on interesting and challenging problems, which are highly relevant to real-life use cases.

- The paper is novel in its framing of the problem - in particular, considering model editing for data removal is fairly unique in literature.

- The paper is quite broad spanning, presenting both multiple attacks and defenses for machine unlearning.

**Weaknesses:**

**Lack of experiments** - The paper could be more comprehensive if more scenarios were presented in the paper, e.g. combining attacks (to maximize extraction under a budget) or combining defenses (to explore how much risks could be minimized in practice). I find it hard to draw actionable conclusions from the results, and further discussion and variety of results may help justify the strength of model editing as the de-facto paradigm for unlearning.

**No comparison to prior works** - I’m concerned about the lack of comparison to prior works. While the paper claims that prior works is not applicable due to focus on removal of influence of a pair $(x,y)$, the problem formalized in the paper (based on the ASR metric) is exactly the recovery of a single token label $y_i$ given a specific prompt $x_i$. Hence, I’m confused why it’s not possible to compare with prior approaches in approximate unlearning.

**Questions:**

- I’m a bit confused by the distinction between the password attempts and parallel extractions scenarios in the context of the paper. Is the distinction here meant to be that information gained in the recovery of a previous attempt may be utilized to aid recovery of the next attempt? If so, I do not understand how the attacks as described utilize information in this way (i.e. it seems all attack methods generate a candidate set for a budget $B$ in parallel anyways). Is such an attack possible?

- In Sec. 2 “Attacking LLMs for Sensitive Information”, the paper claims that the described method does not assume access to the exact text used during pre-training of the model. However, the experiments do assume such access, as even in the rephrasing attack, the ground-truth prompt is perturbed by a rephrasing model. Is it possible to run the rephrasing attack without the ground-truth prompt?

---

> ### Author Response · Authors · 2023-11-17
> **Reply to Reviewer 174w**
>
> > I find it hard to draw actionable conclusions from the results, and further discussion and variety of results may help justify the strength of model editing as the de-facto paradigm for unlearning.
>
> We summarize the actionable conclusions below based on the results in Sec 7.2:
> - *Whitebox attacks*. Max Entropy defense helps best against white-box attacks, which is a positive and directly implementable in practical efforts in sensitive information deletion.
> - *Blackbox attacks*. Although Error injection and Fact erasure defenses are somewhat effective against blackbox attacks, the lack of robustness of the deletion methods to paraphrases of the input is evident from the high attack success rates. Here we note that, like past work with jailbreaks for LLMs, sensitive information deletion is another example of a security issue with LLMs that we are highlighting. They are just not secure against paraphrase attacks.
>
>
> > No comparison to prior works - I’m concerned about the lack of comparison to prior works.
>
> In essence, constrained finetuning is similar to the most common approach for unlearning, which is EWC (Elastic Weight Consolidation). EWC is being used as recently as 2022 in unlearning papers [1]. In the original draft of the paper, we distinguished model editing from unlearning based on whether the goal was to unlearn y|x vs. (x,y). However, the optimization approaches they employ are actually quite similar (specialized and constrained forms of finetuning), and unlearning methods are readily adapted to the problem of sensitive information deletion.
>
> In summary, we believe our constrained finetuning approach (See B.3 and Table 5 in appendix) is already a very comparable baseline to existing unlearning approaches like EWC, and since ROME and MEMIT are known to far outperform constrained finetuning, we believe that we have covered all the relevant baselines and are using the best possible methods for the problem at hand. Although if you know of another unlearning approach that is more qualitatively distinct, we would also be happy to add that.
>
> > I’m a bit confused by the distinction between the password attempts and parallel extractions scenarios in the context of the paper. Is the distinction here meant to be that information gained in the recovery of a previous attempt may be utilized to aid recovery of the next attempt?
>
> To clarify, we tried to give multiple plausible threat models that all map onto the same final Attack Success Metric, in order to more robustly motivate the attack metric we use as being widely relevant to real-world attack scenarios. All experiments utilize the same Attack Success Metric, and there is no “information gained” between different attack attempts in our experiments.
>
> > If so, I do not understand how the attacks as described utilize information in this way (i.e. it seems all attack methods generate a candidate set for a budget B in parallel anyways). Is such an attack possible?
>
> To answer your question (even though it is not the case that our attacks utilize information from past attempts), it would be possible for an attack to leverage past attempts if there was some prior knowledge about candidate correctness being correlated or not. For instance, if it was known that a personal phone number began with either 919 or 336, then one could rule out half the possible candidates by first checking a candidate beginning with 919.
>
> > the paper claims that the described method does not assume access to the exact text used during pre-training of the model. However, the experiments do assume such access, as even in the rephrasing attack, the ground-truth prompt is perturbed by a rephrasing model
>
> In our experiments, we do not use any “ground-truth” prompt from the pretraining data. We assume access to a textual representation of a question that the attacker is interested in knowing the answer to, and we rely on popular model editing datasets CounterFact and ZSRE for these textual examples (Sec 3.1, Sec 6). But CounterFact and ZSRE are not the pretraining datasets that GPT-J and Llama-2-7b were trained on. If we want to know whether GPT-J memorized what country the University of Madrid is in, we actually have no idea what would be the exact text from the pretraining data that prefaces the answer to this question.
>
> In other words, an attacker only needs to know how to ask for some sensitive information, they do not need to know exactly what text prefaced that sensitive information in the model’s pretraining data.
>
> [1] Tanno, Ryutaro, et al. "Repairing Neural Networks by Leaving the Right Past Behind." Advances in Neural Information Processing Systems 35 (2022): 13132-13145.

---

### Author Response · Authors · 2023-11-17
**General Response**

We thank the reviewers for their time, and we appreciate that reviewers found the paper to be “well written” (KWLr) and “novel in its framing of the problem” (174w) with a “well-motivated threat model” (X7HP), and that it “elaborates on the security issues of LLMs” (GnJt) via a “quite thorough” evaluation using “multiple defenses against multiple attacks” (KWLr). We will now address individual questions from each reviewer below.

---

> ### Author Response · Authors · 2023-11-21
>
> We would like to express our gratitude to all the reviewers for their time and effort in reviewing our paper. This serves as a friendly reminder that the deadline for the discussion phase is rapidly approaching. We kindly request your feedback regarding whether our response adequately addressed your concerns. Additionally, if you have any further inquiries or require clarification on any aspects of our paper, we are more than willing to assist before the deadline. Once again, thank you for your valuable inputs and consideration.

---

### Meta-Review · Area_Chair_4dex · 2023-12-08

**Metareview:**

This paper shows that common model editing techniques struggle to remove facts in a way that resists adversarial attempt at recovering them.
Then, the paper introduces additional defense ideas but finds that no defense work universally.

I think this is an important question to study, and the paper does a good job in showing that current fact-removal approaches are not suited for private data.

**Justification For Why Not Higher Score:**

The paper addresses an interesting question and has nice results.
The technical novelty is maybe a bit limited for an oral. The observation is nice but I don't know how much people will be able to build on this result.

**Justification For Why Not Lower Score:**

The paper definitely deserves to be accepted, and touches on a topic that is likely interesting to a broad audience.

---

### Decision · Program_Chairs · 2024-01-16

Accept (spotlight)